# Efficacy and safety of dolutegravir plus emtricitabine versus standard ART for the maintenance of HIV-1 suppression: 48-week results of the factorial, randomized, non-inferiority SIMPL'HIV trial

**Delphine Sculier** [1,2]*, **Gilles Wandeler** [3,4], **Sabine Yerly** [5], **Annalisa Marinosci** [1],
**Marcel Stoeckle** [6], **Enos Bernasconi** [7], **Dominique L. Braun** [8,9], **Pietro Vernazza** [10],
**Matthias Cavassini** [11], **Marta Buzzi** [1], **Karin J. Metzner** [8,9], **Laurent A. Decosterd** [12],
**Huldrych F. Günthard** [8,9], **Patrick Schmid** [10], **Andreas Limacher** [13],
**Matthias Egger** [4,14,15], **Alexandra Calmy** [1,16], and the Swiss HIV Cohort Study (SHCS)[¶]

1 HIV/AIDS Unit, Department of Infectious Diseases, Geneva University Hospitals, Geneva, Switzerland,
2 Private Practice Office, Geneva, Switzerland, 3 Department of Infectious Diseases, Bern University
Hospital, University of Bern, Bern, Switzerland, 4 Institute of Social and Preventive Medicine, University of
Bern, Bern, Switzerland, 5 Laboratory of Virology, Geneva University Hospitals, Geneva, Switzerland,
6 Division of Infectious Diseases and Hospital Epidemiology, University Hospital of Basel, University of Basel,
Switzerland, 7 Service of Infectious Diseases, Lugano Regional Hospital, Lugano, Switzerland, 8 Division of
Infectious Diseases and Hospital Epidemiology, University Hospital of Zurich, University of Zurich, Zurich,
Switzerland, 9 Institute of Medical Virology, University of Zurich, Zurich, Switzerland, 10 Division of Infectious
Diseases and Hospital Epidemiology, Kantonsspital St. Gallen, St. Gallen, Switzerland, 11 Department of
Infectious Diseases, University Hospital of Lausanne, Lausanne, Switzerland, 12 Pharmacology Laboratory,
Clinical Pharmacology Department, University of Lausanne, Lausanne, Switzerland, 13 CTU Bern, University
of Bern, Bern, Switzerland, 14 Centre for Infectious Disease Epidemiology and Research, University of Cape
Town, Cape Town, South Africa, 15 Population Health Sciences, Bristol Medical School, University of Bristol,
Bristol, United Kingdom, 16 Faculty of Medicine, University of Geneva, Geneva, Switzerland

¶ Membership of the Swiss HIV Cohort Study is provided in the Acknowledgments.
* delphine.sculier@hin.ch

journal.pmed.1003421

of Southampton, UNITED KINGDOM

**Data Availability Statement:** Data underlying the
reported findings are provided upon request.

## Abstract

### Background

Dolutegravir (DTG)–based dual therapy is becoming a new paradigm for both the initiation
and maintenance of HIV treatment. The SIMPL'HIV study investigated the outcomes of viro-
logically suppressed patients on standard combination antiretroviral therapy (cART) switch-
ing to DTG + emtricitabine (FTC). We present the 48-week efficacy and safety data on
DTG + FTC versus cART.

### Methods and findings

SIMPL'HIV was a multicenter, open-label, non-inferiority randomized trial with a factorial
design among treatment-experienced people with HIV in Switzerland. Participants were
enrolled between 12 May 2017 and 30 May 2018. Patients virologically suppressed for at
least 24 weeks on standard cART were randomized 1:1 to switching to DTG + FTC or to
continuing cART, and 1:1 to simplified patient-centered monitoring versus standard

Instructions with contact information as well as further relevant documents are available at the Bern Open Repository (BORIS): https://boris.unibe.ch/id/eprint/146808.

**Funding:** This study has been financed within the framework of the Swiss National Science Foundation (grants 166819, 17481) and the Swiss HIV Cohort Study, supported by the Swiss National Science Foundation (grant #177499), by SHCS project #826 and by the SHCS research foundation. The data are gathered by the Five Swiss University Hospitals, two Cantonal Hospitals, 15 affiliated hospitals and 36 private physicians (listed in http://www.shcs.ch/180-health-care-providers). LD's project is funded by the SHCS. AC is Sponsor and Principal Investigator. The Funder did not play any role in the study design, data collection and analysis, decision to publish, or preparation of the manuscript.

**Competing interests:** I have read the journal's policy and the authors of this manuscript have the following competing interests: KJM has received travel grants and honoraria from Gilead Sciences, Roche Diagnostics, GlaxoSmithKline, Merck Sharp & Dohme, Bristol-Myers Squibb, ViiV and Abbott; and the University of Zurich received research grants from Gilead Science, Roche, and Merck Sharp & Dohme for studies that Dr Metzner serves as principal investigator, and advisory board honoraria from Gilead Sciences. AC is a member of the WHO guidelines panel (GSG) from 2017 up to 2020. The HIV Unit of Geneva University Hospitals (HUG) received unrestricted educational grants from ViiV, AbbVie, Gilead and MSD; ViiV, Gilead and MSD also provided financial support to the LIPO group and metabolism (day hospital) in the HIV/AIDS Unit of HUG. HFG has received unrestricted Research Grants from Gilead Sciences and Roche; fees for data and safety monitoring board membership from Merck; consulting/advisory board membership fees from Gilead Sciences, ViiV, Merck, Sandoz and Mepha. MS has received advisory board honoraria (paid to the institution) from Gilead, MSD Janssen, ViiV, Sandoz and Mepha and travel/conference Grants from Gilead and MSD. MC institution received Research Grants from Gilead and Viiv, and expert opinion's fees from Abbvie, Gilead, MSD, Viiv and Sandoz. DLB has received honoraria and travel grants from Merck, Gilead and ViiV. GW has received research grants from Gilead Sciences and travel grants/honoraria from Gilead Sciences, AbbVie and ViiV, (paid to the institution). ME is a member of the Editorial Board of Plos Medicine. DS, AM, MB, ME, SY, LD, AL, PS, EB and PV have no competing of interests to declare.

monitoring. The primary endpoint was the proportion of patients virologically suppressed with <100 copies/ml through 48 weeks. The secondary endpoints included virological suppression at 48 weeks according to the US Food and Drug Administration (FDA) snapshot analysis. Non-inferiority of DTG + FTC versus cART for viral suppression was assessed using a stratified Mantel–Haenszel risk difference, with non-inferiority declared if the lower bound of the 95% confidence interval was greater than −12%. Adverse events were monitored to assess safety. Quality of life was evaluated using the PROQOL-HIV questionnaire. Ninety-three participants were randomized to DTG + FTC, and 94 individuals to cART. Median nadir CD4 count was 246 cells/mm$^3$; median age was 48 years; 17% of participants were female. DTG + FTC was non-inferior to cART. The proportion of patients with viral suppression (<100 copies/ml) through 48 weeks was 93.5% in the DTG + FTC arm and 94.7% in the cART arm in the intention-to-treat population (risk difference −1.2%; 95% CI −7.8% to 5.6%). Per-protocol analysis showed similar results, with viral suppression in 96.5% of patients in both arms (risk difference 0.0%; 95% CI −5.6% to 5.5%). There was no relevant interaction between the type of treatment and monitoring (interaction ratio 0.98; 95% CI 0.85 to 1.13; $p = 0.81$). Using the FDA snapshot algorithm, 84/93 (90.3%) participants in the DTG + FTC arm had an HIV-1 RNA viral load of <50 copies/ml compared to 86/94 (91.5%) participants on standard cART (risk difference −1.1%; 95% CI −9.3% to 7.1%; $p = 0.791$). The overall proportion of patients with adverse events and discontinuations did not differ by randomization arm. The proportion of patients with serious adverse events was higher in the cART arm (16%) compared to the DTG + FTC arm (6.5%) ($p = 0.041$), but none was considered to be related to the study medication. Quality of life improved more between baseline and week 48 in the DTG + FTC compared to the cART arm (adjusted difference +2.6; 95% CI +0.4 to +4.7). The study's main limitations included a rather small proportion of women included, the open label design, and its short duration.

**Conclusions**

In this study, DTG + FTC as maintenance therapy was non-inferior to cART in terms of efficacy, with a similar safety profile and a greater improvement in quality of life, thus expanding the offer of 2-drug simplification options among virologically suppressed individuals.

**Trial registration**

ClinicalTrials.gov NCT03160105.

## Author summary

### Why was the study done?

- Treatment simplification among people with HIV has been tested worldwide and includes reducing the number and/or dosage of antiretroviral drugs, and simplifying monitoring, but without compromising adherence and quality of life.

**Abbreviations:** 3TC, lamivudine; cART, combination antiretroviral therapy; DTG, dolutegravir; EACS, European AIDS Clinical Society; FDA, US Food and Drug Administration; FTC, emtricitabine; InSTI, integrase strand transfer inhibitor; ITT, intention-to-treat; NRTI, nucleoside reverse transcriptase inhibitor; PCM, patient-centered monitoring; PP, per protocol; PWHIV, people with HIV; QoL, quality of life; SHCS, Swiss HIV Cohort Study; SM, standard monitoring.

- We evaluated the combination of dolutegravir (DTG) + emtricitabine (FTC) as an alternative to standard therapy combinations in an HIV population highly representative of routine clinical conditions.
- Simplified patient-centered monitoring was also evaluated, to allow patients to receive care outside the hospital as is usual in Switzerland.

## What did the researchers do and find?

- We conducted the SIMPL'HIV study, a randomized trial with a factorial design, to permit patients with a suppressed viral load to receive a simplified treatment of DTG + FTC and/or simplified patient-centered monitoring.
- Efficacy of the DTG + FTC regimen was defined as keeping the patient's HIV viral load below 100 copies/ml through 48 weeks of study duration.
- The combination of DTG with FTC had a similar efficacy in maintaining viral suppression through 48 weeks of treatment as standard combined antiretroviral therapy, without jeopardizing safety, and it improved patient quality of life.

## What do these findings mean?

- These results provide further evidence on the efficacy of 2-drug DTG-based regimens as a simplified switch option for patients whose viral load is well controlled on standard treatment, including those with a low CD4 nadir, and expands the currently existing options for dual maintenance therapy.
- Our efficacy definition allowed us to observe the occurrence of HIV viral load blips, irrespective of the treatment arm, without any consequences for HIV management.
- The quality of life of the participants was already very satisfactory at the start of the study and further increased over time with the simplified DTG + FTC treatment.

## Introduction

Treatment optimization has been at the forefront of innovative HIV-1 care management. Dual-therapy regimens are already considered appropriate alternatives to standard combination antiretroviral therapy (cART) as they have been approved by the US Food and Drug Administration (FDA) and the European AIDS Clinical Society (EACS) guidelines for both initial and maintenance therapy [1,2].

   Dual therapy may also be associated with reduced toxicity and costs and improved tolerability, adherence, and quality of life (QoL) [3].

   In recent years, there has been a focus on dolutegravir (DTG)–based simplification therapy. DTG is a potent second-generation integrase strand transfer inhibitor (InSTI) with a high genetic barrier to resistance, rare severe side effects, and very few drug–drug interactions [4]. While DTG is associated with a substantial risk of virological failure in maintenance monotherapy [5,6], it remains a good candidate for dual therapy. The GEMINI phase III trials established the non-inferiority and tolerability of DTG plus lamivudine (3TC) in

ART-naïve patients [7]. In selected patients, the TANGO study demonstrated the non-inferiority and safety of DTG plus 3TC in maintenance therapy versus a standard cART containing tenofovir alafenamide [8]. In heavily treatment-experienced patients and in the presence of historical 3TC resistance, the dual regimen DTG + 3TC allowed maintenance of virological suppression at week 48, as shown by the DOLULAM and ART-PRO pilot clinical trials [9,10].

We aimed to investigate the efficacy and safety of DTG plus emtricitabine (FTC) dual therapy, which has not yet been examined to our knowledge. FTC is similar to 3TC with respect to the convenience, safety, and resistance profile, but it has a longer intracellular half-life activity and shows greater in vitro potency, results that contributed to the selection of FTC for subsequently developed triple antiretroviral fixed dose combinations [11]. We aimed to examine the efficacy and safety of DTG + FTC dual therapy in a nationally representative sample of people with HIV (PWHIV) that included a substantial percentage of women. We did not use any CD4 nadir or HIV-1 RNA zenith restrictions, thus mirroring routine care while reducing the number of drugs and costs to the health system. Finally, no trial to our knowledge has simultaneously explored the non-inferiority of dual therapy and of simplified and individualized patient-centered treatment monitoring, which was made possible by a factorial study design.

SIMPL'HIV was a multicenter, non-inferiority, open-label, randomized, factorial trial conducted in Switzerland. The primary objective of SIMPL'HIV was to assess the 48-week efficacy and safety of DTG + FTC dual maintenance therapy in virologically suppressed PWHIV compared with standard cART. As a co-primary objective, the study aimed to compare a patient-centered monitoring (PCM) approach with standard 3-monthly routine surveillance (standard monitoring [SM]). Here we focus on the comparison between DTG + FTC and cART in terms of efficacy, safety, and QoL; costs and other outcomes for the comparison of the monitoring options are not reported in this paper.

## Methods

### Design and participants

SIMPL'HIV was performed in the 7 main Swiss HIV Cohort Study (SHCS) sites [12] with participants enrolled between 12 May 2017 and 30 May 2018. The week 48 database was frozen on 3 July 2019. The study, with an open-label design, recruited PWHIV aged 18 years or over and already enrolled in the SHCS network. Patients were eligible if they were on any EACS-recommended cART [13] and virologically suppressed for at least 24 weeks prior to enrollment, with virological suppression defined as HIV-1 RNA < 50 copies/ml, although we allowed a single blip of <200 copies/ml in the past 6 months. We excluded patients who had previous treatment change due to an unsatisfactory virological response such as slow initial or incomplete virological suppression or rebound. Patients with only a transmitted M184V mutation became eligible from 27 June 2017, following a protocol amendment, in an effort to include individuals with transmitted nucleoside reverse transcriptase inhibitor (NRTI)–selected mutations. Exclusion criteria included the following: presence of InSTI resistance mutations according to the French National Agency for AIDS Research version 29 algorithm [14], pregnant or breastfeeding women, HIV-2 infection, creatinine clearance of <50 ml/min, transaminase elevation>2.5 times the upper limit of the norm, known hypersensitivity to DTG or FTC, known or suspected non-adherence to current treatment, and evidence of acute or chronic hepatitis B virus infection. Written informed consent was obtained from each participant before the initiation of study procedures.

## Randomization

Participants were randomly assigned 1:1 to switch to DTG + FTC dual maintenance therapy or continue their cART, and 1:1 to PCM or SM (S1 Fig). An independent statistician generated a computer-based random allocation sequence stratified by study site, using randomly permutated blocks of size 4 and 8 to randomize patients to the 4 arms. The randomization list was implemented by an independent data manager in a web-based data management system in order to ensure concealment of allocation.

## Procedures

Patients were randomized to receive a once-daily 2-drug regimen of DTG 50 mg plus FTC 200 mg or to continue their standard cART, as well as to PCM or SM. Allocation to SM consisted of 3-monthly routine immunological and blood safety tests including CD4 cell count, lipid profile, glucose level, renal and hepatic function tests, and creatinine kinase, with all visits and laboratory analyses conducted at the affiliated SHCS sites. Participants allocated to the PCM arm had a restricted immunological and blood safety monitoring at weeks 0 and 48. In addition, participants were offered the choice to complete some of the study visits by a telephone call with a study nurse, to have their drugs delivered to a specified address, or to have their blood tests performed at alternative locations, including certified private laboratories.

All participants had HIV-1 RNA measurements performed at all study visits, which were scheduled every 6 weeks between baseline and week 12, and every 12 weeks thereafter until the study ended. We quantified HIV-1 RNA levels using PCR assays with a limit of detection of 20 copies/ml or less at local virology laboratories. Patients with HIV-1 RNA between 20 and 99 copies/ml were retested after 6 weeks (±5 days). Patients with HIV-1 RNA $\geq$ 100 copies/ml were retested after 14 days (±5 days), received additional adherence sessions, had their concomitant medications reviewed, and had drug plasma concentration measurements performed. In patients with virological failure, defined as 2 consecutive plasma HIV-1 RNA values $\geq$ 100 copies/ml, genotypic resistance testing was performed as part of routine care by 4 laboratories in Switzerland authorized by the Federal Office of Public Health. All laboratories performed population-based sequencing. Drug resistance was defined using the French National Agency for AIDS Research version 29 algorithm [14]. If randomized to the DTG +FTC dual therapy arm, patients were switched to a standard cART regimen while awaiting results of genotypic resistance testing. Patients in the control arm remained on their cART regimen while awaiting the genotypic results. Treatment was adjusted in both cases according to genotypic resistance results. We quantified the plasma levels of DTG and FTC by liquid chromatography coupled to tandem mass spectrometry in the Laboratory of Clinical Pharmacology at Lausanne University Hospital at week 48, using adaptations of previously published methods [15–17].

We recorded adverse events, concomitant medications, and adherence checks and performed symptom-directed physical examinations at all study visits. We graded adverse events by severity on a scale from 1 to 4 using the categories in the International Council for Harmonisation of Technical Requirements for Registration of Pharmaceuticals for Human Use E2A guidelines [18]. Serious adverse events were defined according to the same guidelines. Urinary pregnancy testing was done at the screening visit and every 6 months thereafter (i.e., at week 24 and 48 visits) for women of childbearing age.

## Outcomes

The primary outcome for the comparison between DTG + FTC and cART was the maintenance of HIV-1 RNA < 100 copies/ml through the 48-week duration of the study. A single blip, defined as HIV-1 RNA between 100 and 199 copies/ml at any time during the study

period, was allowed as long as this measurement was followed by an HIV-1 RNA result of <50 copies/ml. Secondary endpoints included the proportion of patients with HIV-1 RNA < 50 copies/ml at week 48 according to the FDA snapshot algorithm [19], the proportion with HIV-1 RNA < 50 copies/ml through 48 weeks, the proportion experiencing loss of future drug options and time to loss of virological response, changes in CD4 cell count, changes in HIV DNA, adherence through the study, and changes in health-related QoL as assessed by the PROQOL-HIV questionnaire [20,21] from baseline to week 48. We also assessed on which drug regimen participants chose to stay at study termination. To be aligned with FDA guidelines, the protocol was amended on 30 July 2019 to increase the window for the FDA snapshot analysis from 7 to 21 days.

Safety endpoints included the incidence, type, and seriousness of adverse events, as well as the discontinuation of treatment. Renal function, lipids, Framingham cardiovascular risk score, glucose, and weight were monitored by assessing changes from baseline to week 48.

The co-primary outcome, for the comparison between PSM and SM, was the direct costs from the healthcare system perspective of the 2 monitoring options. Secondary outcomes for the comparison of monitoring arms included cost-effectiveness; the number of extra visits during the study period; in the PSM arm, participants' monitoring satisfaction and the proportion of participants expressing willingness to change monitoring options during the study; and all participants' satisfaction with treatment, monitoring, and study participation. Primary and secondary outcomes related to the comparison of the monitoring options will not be presented in this paper.

## Sample size calculation and statistical analyses

We determined the target sample size for the non-inferiority comparison between DTG +FTC and cART on the primary outcome, assuming that there was no interaction between the treatment and monitoring strategy. The targeted sample size of 184 patients (allowing for at least 10% loss to follow-up) was based on 80% power, a 2.5% 1-sided alpha level, expected proportions of HIV-1 RNA < 100 copies/ml through 48 weeks of 94% and 97% in the DTG + FTC and cART arm, respectively, and a non-inferiority margin of −12%, as used in previous non-inferiority HIV trials [22–29].

We primarily analyzed the trial as a stratified 2-arm trial comparing DTG + FTC with cART, with monitoring type (PCM versus SM) being the stratification factor. The primary outcome was assessed in both the intention-to-treat (ITT) and per-protocol (PP) populations, as recommended by the FDA guidance for non-inferiority trials [30]. PP exclusion criteria included missing HIV-1 RNA results at week 48 (±21-day window), unless virological failure occurred before; treatment adherence < 80%; and any major protocol deviations. We assessed interaction between the type of drug treatment and monitoring by calculating an interaction ratio [31] and a p-value using the Mantel–Haenszel test of homogeneity in order to exclude that the treatment effect on viral suppression was different depending on the type of monitoring.

We summarized baseline characteristics and adverse events using descriptive statistics. We evaluated the primary outcome using the Mantel–Haenszel risk difference stratified by the monitoring type [32], declaring non-inferiority if the corresponding lower 95% confidence limit was above the margin of −12%. We compared secondary binary outcomes using the Cochran–Mantel–Haenszel test statistic and Mantel–Haenszel risk difference, stratified by the type of monitoring. We analyzed continuous variables using linear regression adjusted for the type of monitoring and the baseline value. All statistical analyses were performed with R software, version 3.6.1 [33].

An independent data monitoring committee reviewed descriptive safety data after the first 100 participants had completed the week 24 study visit. SIMPL'HIV is registered on ClinicalTrials.gov (NCT03160105). This report was written according to the CONSORT statement for non-inferiority trials; the CONSORT checklist is available in S1 Text [34].

### Ethics statement

The study protocol was approved by both the leading and local ethics committees in Switzerland, in accordance with the Helsinki Declaration and good clinical practice. The full protocol is available in S2 Text.

## Results

Of 873 individuals screened for eligibility, 188 were randomly assigned either to switch to DTG + FTC dual therapy or to continue their cART (Fig 1). One randomization occurred by mistake (ineligible patient mistakenly randomized), leading to 93 participants allocated to

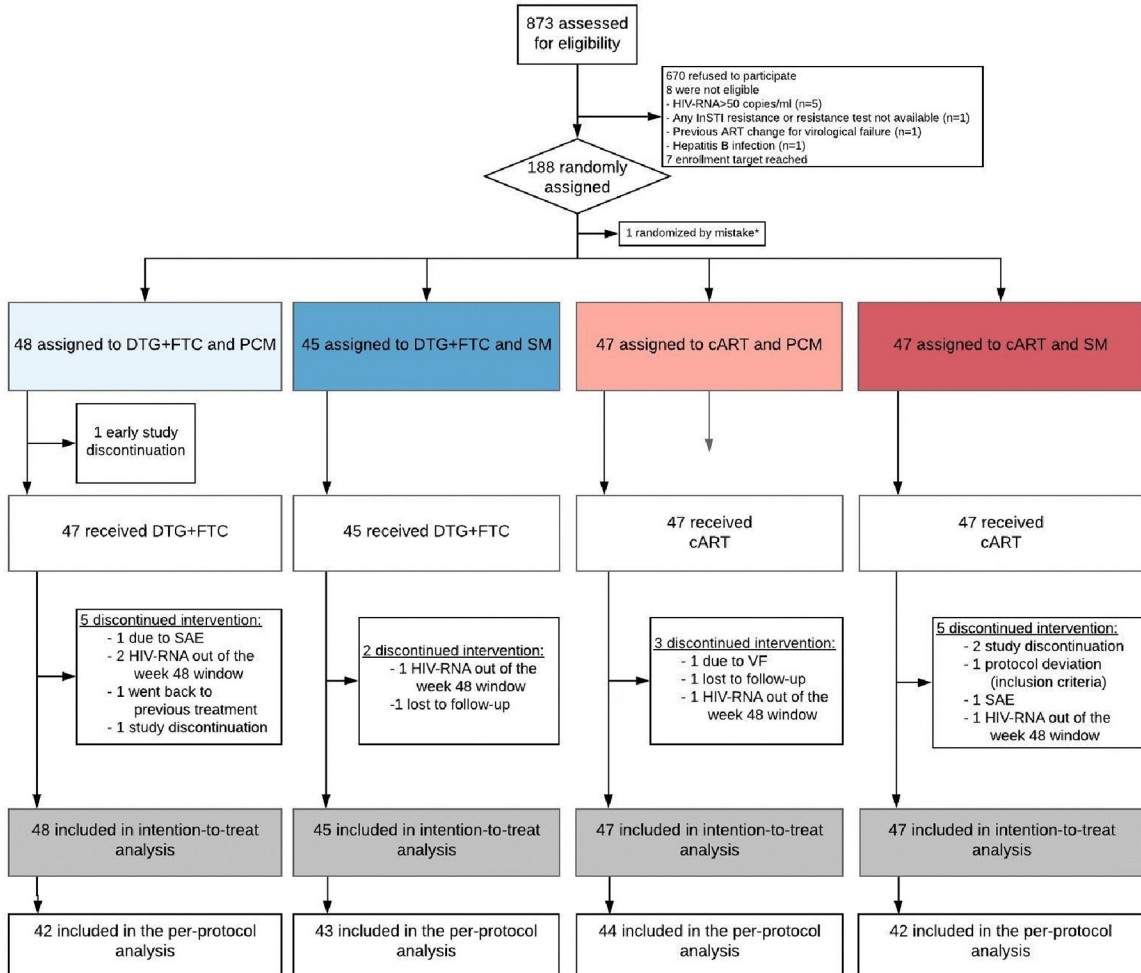

**Fig 1. Overview of study flow chart.** *Ineligible patient mistakenly randomized. cART, combination antiretroviral therapy; DTG, dolutegravir; FTC, emtricitabine; InSTI, integrase strand transfer inhibitor; PCM, patient-centered monitoring; SAE, serious adverse event; SM, standard monitoring; VF, virological failure.

DTG + FTC dual therapy, with 48 participants in the PCM arm and 45 in the SM arm, and 94 participants allocated to continuing cART, with 47 in each of the monitoring arms. The number of participants who discontinued treatment before week 48 was similar between the 2 treatment arms, 8 in each arm. Reasons for study discontinuation included premature stop or loss to follow-up ($n = 6$), discontinuation due to an adverse event ($n = 2$), change to previous cART for a reason other than virological failure ($n = 1$), and low adherence ($n = 1$).

## Baseline characteristics

Key demographic and baseline characteristics were well balanced between the treatment arms in the ITT population (Table 1). Participants were mostly white (79%), with a median age of 48 years (interquartile range [IQR] 40–54); 17% were female. The median CD4 nadir was 246 cells/mm$^3$ (IQR 103–330) in the DTG + FTC arm, and 253 cells/mm$^3$ (IQR 174–367) in the cART arm. Standard cART at inclusion comprised an association of 2 NRTIs with an InSTI for two-thirds of participants. DTG was the InSTI received before study entry for one-half of all participants.

## Primary outcome (efficacy)

DTG was non-inferior to cART in maintaining viral suppression; 87/93 (93.5%) participants in the DTG + FTC dual therapy arm and 89/94 (94.7%) in the cART arm achieved HIV-1

**Table 1. Demographic and clinical baseline characteristics of the intention-to-treat population.**

| Characteristic | DTG + FTC ($n = 93$) | cART ($n = 94$) |
|---|---|---|
| **Median age (years) [IQR]** | 48 [40–53] | 47 [42–54] |
| **Female, $n$ (%)** | 14 (15) | 18 (19) |
| **Ethnicity, $n$ (%)** | | |
| White | 73 (78.5) | 75 (79.8) |
| Asian | 2 (2.2) | 3 (3.2) |
| Black | 12 (12.9) | 12 (12.8) |
| Other | 6 (6.4) | 4 (4.2) |
| **Positive HCV serology, $n$ (%)** | 8 (8.6) | 9 (9.6) |
| **Median CD4 baseline (cells/mm$^3$) [IQR]** | 664 [515–898] | 686 [497–847] |
| **Median CD4 nadir (cells/mm$^3$) [IQR]** | 246 [103–330] | 253 [174–367] |
| **Median viral load zenith (copies/ml) [IQR]** | 109,757 [41,200–326,000] | 100,000 [31,000–303,000] |
| **Median body mass index (kg/m$^2$) [IQR]** | 25 [23–26] | 25 [23–27] |
| **Median time since ART initiation (years) [IQR]** | 7.9 [4.1–12] | 7.1 [3.8–13] |
| **Median time since HIV diagnosis (years) [IQR]** | 10 [5–15] | 11 [6–17] |
| **AIDS events during the last 5 years, $n$ (%)** | 1 (1.1) | 0 |
| **cART at inclusion, $n$ (%)** | | |
| 2 NRTI + InSTI | 59 (63.4) | 62 (66.0) |
| 2 NRTI + 1 NNRTI | 27 (29.0) | 24 (25.5) |
| 2 NRTI + 1 boosted PI | 5 (5.4) | 6 (6.4) |
| Other | 2 (2.2) | 2 (2.1) |
| **Documented M184V mutation, $n$ (%)** | 0 | 1 (1.1) |

AIDS, acquired immunodeficiency syndrome; ART, antiretroviral therapy; cART, combined antiretroviral therapy; DTG, dolutegravir; FTC, emtricitabine; HCV, hepatitis C virus; HIV, human immunodeficiency virus; InSTI, integrase strand transfer inhibitor; IQR, interquartile range; NNTRI, non-nucleoside reverse transcriptase inhibitor; NTRI, nucleoside reverse transcriptase inhibitor; PI, protease inhibitor.

**Table 2. Proportions of patients with HIV-1 RNA < 100 copies/ml throughout 48 weeks.**

| Outcome | DTG + FTC | cART | Difference (95% CI) | Non-inferiority met?[*] |
|---|---|---|---|---|
| **ITT analysis** | n = 93 | n = 94 | | |
| HIV-1 RNA < 100 copies/ml throughout 48 weeks (±21 days) | 93.5% (87) | 94.7% (89) | −1.2% (−7.8% to +5.6%) | Yes |
| **PP analysis** | n = 85 | n = 86 | | |
| HIV-1 RNA < 100 copies/ml throughout 48 weeks (±21 days) | 96.5% (82) | 96.5% (83) | −0.0% (−5.6% to +5.5%) | Yes |

Mantel–Haenszel risk difference with 95% confidence intervals for the ITT and PP analysis.

[*]Non-inferiority-margin: −12%.

cART, combined antiretroviral therapy; CI, confidence interval; DTG, dolutegravir; FTC, emtricitabine; ITT, intention-to-treat; PP, per-protocol.

RNA < 100 copies/ml throughout 48 weeks in the ITT population (risk difference −1.2%; 95% CI −7.8% to 5.6%; non-inferiority margin −12%) (Table 2; Fig 2A). PP analysis showed similar results, with 96.5% of patients maintaining viral suppression in both arms (treatment difference 0.0%; 95% CI −5.6% to 5.5%) (Table 2). There was no relevant interaction between the type of drug treatment and type of monitoring for the primary outcome (interaction ratio 0.98; 95% CI 0.85 to 1.13; $p = 0.81$). We did not observe any difference in efficacy by monitoring arm (S1 Table). One patient (0.5%) in the cART arm on abacavir/3TC/DTG failed according to the protocol virological failure definition, i.e., 2 consecutive HIV-1 RNA values ≥ 100 copies/ml. Five patients presented with a single HIV-1 RNA ≥ 200 copies at least once during the 48-week follow-up, followed by an undetectable viral load, and were not considered as true virological failures.

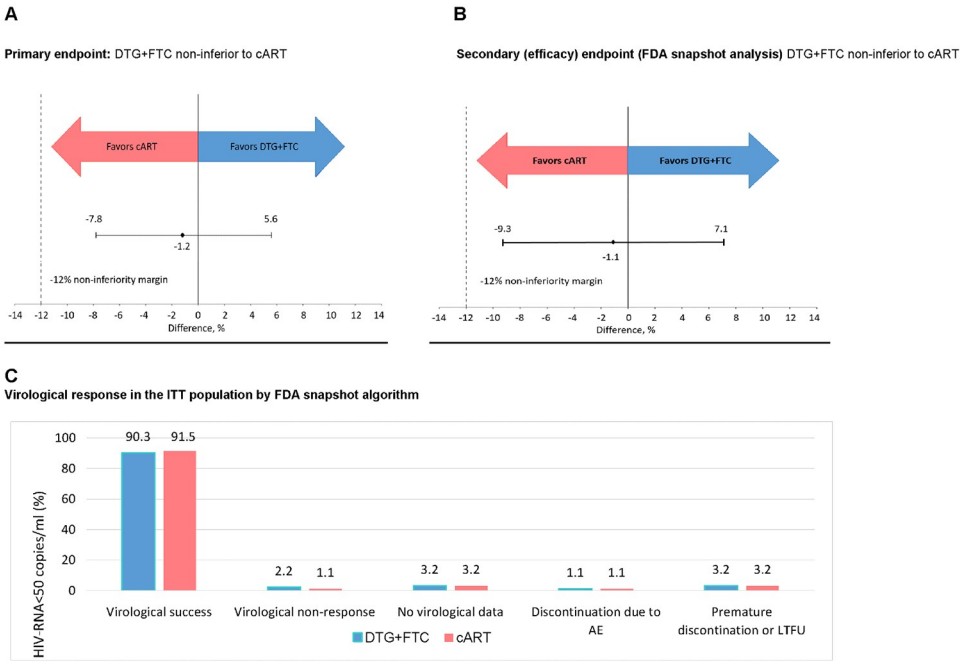

**Fig 2. Virological outcomes at week 48.** Adjusted treatment difference for the primary outcome (A) and the FDA snapshot analysis (B) and virological response in the ITT population by the FDA snapshot algorithm (C). AE, adverse event; cART, combined antiretroviral therapy; CI, confidence interval; DTG, dolutegravir; FDA, US Food and Drug Administration; FTC, emtricitabine; ITT, intention-to-treat; LTFU, lost to follow-up.

## Secondary outcomes

Using the FDA snapshot algorithm, 84/93 (90.3%) participants in the DTG + FTC arm had HIV-1 RNA < 50 copies/ml at week 48, compared to 86/94 (91.5%) in the standard cART arm (risk difference −1.1%; 95% CI −9.3% to 7.1%; $p$ = 0.791) (Fig 2B and 2C). Two patients, 1 in each arm, had HIV-1 RNA ≥ 50 copies/ml at week 48. These patients did not overlap with the participant who virologically failed according to the primary endpoint. For 5 additional patients, week 48 HIV-1 RNA measurements were available but were outside the window period of ±21 days (S2 Table). Adherence through 48 week reached 98.6% in the DTG + FTC arm and 98.7% in the cART arm (risk difference +0.2%; 95% CI −0.9% to 1.4%; $p$ = 0.671).

We did not observe any new occurrence of resistance, and thus no loss of future drug options, among patients who virologically failed according to the primary or secondary outcome. Of note, we did not detect DTG plasma levels for the patient who experienced virological failure according to the primary outcome. Still, 41L and 69D thymidine analogue mutations were present at the time of virological failure. These mutations were transmitted as they were observed in the genotyping performed at the time of HIV diagnosis, before any exposure to HIV treatment. Overall, no patient required a change of therapy, although 1 patient in the DTG + FTC arm requested to switch back to standard cART due to low-level viremia occurring before week 48. As we only observed 1 virological failure, we were not able to explore risk factors for this outcome.

## Adverse effects

The most commonly reported adverse events across both treatment arms were upper respiratory infection (20.9%), headache (10.2%), diarrhea (7%), and insomnia (4.3%) (Table 3). The proportion of patients with serious adverse events was higher in the cART arm (16%) compared to the DTG + FTC arm (6.5%) (risk difference −9.5%; 95% CI −18.2% to −0.4%; $p$ = 0.041). None of the serious adverse events was considered to be related to the study medication. There was no difference in the proportion of participants with serious adverse events by monitoring arm (10.5% in the PCM arm versus 12% in the SM arm; risk difference −1.3% (95% CI −10.2% to +7.7%; $p$ = 0.781) (S3 Table). One participant in each arm (1%) had an adverse event leading to discontinuation of the study drug (suicidal ideation in the DTG +

**Table 3. Summary of reported adverse events.**

| Adverse event outcome | DTG + FTC (*n* = 93) | cART (*n* = 94) |
|---|---|---|
| **At least 1 adverse event, *n* (%)** | 65 (69.9) | 57 (60.6) |
| Upper respiratory infection | 20 (21.5) | 19 (20.2) |
| Diarrhea | 12 (12.9) | 1 (1.1) |
| Headache | 10 (10.8) | 9 (9.6) |
| Insomnia | 6 (6.5) | 2 (2.1) |
| Fever | 4 (4.3) | 5 (5.3) |
| Myalgia | 4 (4.3) | 2 (2.1) |
| Rash | 3 (3.2) | 6 (6.4) |
| **At least 1 serious adverse event, *n* (%)** | 6 (6.5) | 15 (16) |
| **Adverse event leading to withdrawal from the intervention, *n* (%)** | 1 (1.1) | 1 (1.1) |
| **Adverse event leading to withdrawal from the study, *n* (%)** | 0 | 0 |

cART, combined antiretroviral therapy; DTG, dolutegravir; FTC, emtricitabine.

**Table 4. Changes in safety endpoints between baseline and week 48.**

| Safety endpoint | Mean (±SD) change between baseline and week 48 | | Adjusted difference (95% CI) | p-Value |
|---|---|---|---|---|
| | DTG + FTC (n = 93) | cART (n = 94) | | |
| CD4 cell count, cells/mm$^3$ | +10.8 (±174.4) | −9.4 (±156.6) | +23.1 (−24.0; +70.2) | 0.334 |
| Total cholesterol, mmol/l | −0.24 (±0.63) | −0.13 (±0.69) | −0.06 (−0.24; +0.11) | 0.484 |
| LDL-cholesterol, mmol/l | −0.16 (±0.57) | +0.01 (±0.59) | −0.14 (−0.30; +0.02) | 0.095 |
| Triglycerides, mmol/l | −0.11 (±0.92) | −0.14 (±0.78) | −0.00 (−0.21; +0.22) | 0.983 |
| Estimated creatinine clearance (CKD-EPI), ml/min/1.73 m$^2$ | −2.4 (±11.8) | +1.1 (±9.5) | −4.3 (−7.3; −1.3) | 0.006 |
| Weight, kg | +1.3 (±3.3) | +0.3 (±3.6) | +1.0 (−0.0; +2.0) | 0.058 |
| Glucose profile in mmol/l | −0.2 (±1.2) | +0.1 (±1.2) | −0.3 (−0.6; −0.0) | 0.047 |
| Framingham-calculated cardiovascular risk | +0.1 (±2.5) | +0.4 (±2.6) | −0.3 (−1.0; +0.4) | 0.434 |

cART, combined antiretroviral therapy; CI, confidence interval; CKD-EPI, Chronic Kidney Disease Epidemiology Collaboration; DTG, dolutegravir; FTC, emtricitabine; LDL, low-density lipoprotein.

FTC arm and arthralgia in the cART arm). No participant became pregnant during the study. The full list of the reported serious adverse events is presented in S4 Table.

We observed no difference in the change in CD4 count or lipid profile from baseline to week 48 between the treatment arms. Creatinine clearance was significantly lower in the DTG + FTC arm compared to the cART arm (adjusted difference −4.3 ml/min/1.73 m$^2$; 95% CI −7.3 to −1.3; $p = 0.006$). We observed a slight increase in weight in both arms, but without a statistically significant difference between the arms. Changes between baseline and week 48 in safety endpoints are presented in Table 4.

## Patient-reported outcomes

Mean scores for QoL at baseline and week 48 were high in both treatment arms as measured by the PROQOL-HIV questionnaire. The PROQOL score was rated above 80/100 points for all participants at week 48. Change in QoL between baseline and week 48 with this measure was statistically superior in the DTG + FTC arm (+2.9 points) compared to the cART arm (+0.3 points) (adjusted difference +2.6, 95% CI +0.4 to +4.7; $p = 0.023$) (Table 5). We observed no significant difference in change in QoL in patients randomized in the PCM arm (+2.2 points) compared to the SM arm (+1 point) (adjusted difference +1.6; 95% CI −0.6 to 3.8; $p = 0.166$).

At study termination, patients were offered the choice of remaining on their allocated treatment or switching to another regimen; 85.6% of patients in the dual therapy arm decided to stay on the dual combination of DTG + FTC, whereas 32.2% of participants in the cART decided to initiate DTG + FTC.

**Table 5. Summary of patients' quality of life scores by treatment arm.**

| Outcome measure | Mean (±SD) change; n | | Adjusted difference (95%CI) | p-Value |
|---|---|---|---|---|
| | DTG + FTC (n = 93) | cART (n = 94) | | |
| Change in QoL between baseline and week 12 (PROQOL-HIV questionnaire) | +2.3 (±6.7); n = 79 | +0.2 (±7.0); n = 79 | +2.1 (+0.1; +4.1) | 0.041 |
| Change in QoL between baseline and week 48 (PROQOL-HIV questionnaire) | +2.9 (±6.7); n = 80 | +0.3 (±8.6); n = 85 | +2.6 (+0.4; +4.7) | 0.023 |

cART, combined antiretroviral therapy; DTG, dolutegravir; FTC, emtricitabine; QoL, quality of life.

## Discussion

SIMPL'HIV, a nationwide randomized trial in Switzerland, tested FTC in combination with DTG as a 2-drug regimen. Switching HIV-1-infected virologically suppressed participants to DTG + FTC was non-inferior in terms of efficacy compared to maintaining standard cART through a duration of 48 weeks. ITT analysis showed that the percentage of patients maintaining viral suppression (<100 copies/ml) through 48 weeks was 93.5% in the DTG + FTC arm and 94.7% in the cART arm, thus confirming non-inferiority of the 2-drug arm. We demonstrated a good safety profile of the dual therapy, with few serious adverse events, 6.5% of patients in the DTG + FTC arm and 16% in the cART arm. None was drug-related. QoL improved more between baseline and week 48 in the DTG + FTC arm compared to the cART arm, while there was no difference in QoL over time in the PCM arm compared to the SM one. The lack of improvement in QoL whatever the monitoring arm might reflect the already facilitated access to care in Switzerland and the high level of trust by patients in the healthcare system.

The FDA snapshot virological outcome, with 90.3% of patients having virological suppression in the DTG + FTC arm versus 91.5% in the cART arm, also met the non-inferiority criterion. Our results are similar to those of the TANGO trial, in which 93.2% of patients had virological suppression in the DTG/3TC arm versus 93% in the standard ART arm [8], and the SWORD studies, in which virological suppression was achieved in 95% of participants in both the DTG/rilpivirine and control arms at week 48 [35]. However, the SIMPL'HIV study had fewer eligibility constraints compared to these trials. We had no restriction in the comparator cART arm as long as the treatment followed EACS guidelines. The TANGO trial recruited patients on a tenofovir alafenamide fumarate (TAF)–based regimen, and the SWORD studies included participants on 2 NRTIs plus a third agent, preferably not DTG, before any switch [8,35]. Furthermore, we included patients independent of US Centers for Disease Control and Prevention clinical stage, nadir CD4 count, or HIV-1 zenith viral load values. Indeed, median nadir CD4 count for our participants was quite low (246 cells/mm$^3$ [IQR 147–340]), thus reflecting routine urban HIV care.

The safety profile of DTG + FTC was consistent with the safety profiles of DTG/3TC and DTG/rilpivirine in the TANGO and SWORD trials, respectively. We observed a significantly lower estimated creatinine clearance in the DTG + FTC compared to the cART arm. DTG is known to increase serum creatinine and reduce creatinine clearance by inhibiting the tubular secretion of creatinine [36]. However, we did not measure cystatin C level and therefore cannot judge if this finding is clinically relevant. Similarly to TANGO and SWORD, weight slightly increased in both treatment arms, but the difference between arms was not significant [8,35].

Our novel viral suppression outcome, defined as HIV-1 RNA < 100 copies/ml through 48 weeks' duration, allowed us to observe the occurrence of blips in both treatment arms, with no consequences for HIV management. However, we did not assess inflammatory parameters, which may be affected in the context of continuous antigen exposure. SIMPL'HIV addressed treatment optimization not only by assessing a 2-drug maintenance regimen, but also by simplifying HIV-1 monitoring through reducing the frequency of immunological and safety measurements and providing patient-friendly approaches for follow-up. Treatment optimization cannot only be driven by changes or modulations of antiretroviral drugs; drug and care delivery, including the frequency of laboratory tests, contribute to patient satisfaction and cost control. To our knowledge, no study has assessed such management optimization in a high-income country.

Our study has some limitations. First, despite our efforts to include women in a similar proportion to our national prospective SHCS cohort, i.e., 28%, we failed to reach this target. A qualitative study is planned to understand barriers—a possible implicit sex bias in research both from the researcher side and from the participant side—to the participation of women in clinical trials in Switzerland. Second, the open-label design may have introduced bias regarding the reporting of adverse events. Such bias may explain the lower number of serious adverse events observed in the DTG + FTC arm, because all participants were stable on standard cART for nearly a decade before the start of the study, and such a difference was not expected. Finally, our study duration of 48 weeks was short, and a longer follow-up is needed to document the durability of DTG + FTC dual therapy in terms of efficacy and safety. All study participants will be followed over an extended 3-year period to assess these issues.

The only virological failure occurred in a patient who had transmitted thymidine analogue mutations at the time of HIV-1 diagnosis, before any exposure to antiretroviral drugs. Thus, the results from resistance tests, as well as a detailed medical history, should be considered before switching to a simplified therapy. Of note, this patient was randomized to cART, and we did not detect any drug plasma concentration at the time of virological failure. Importantly, no patient lost drug options during the course of the study.

In conclusion, this study provides evidence that the combination of DTG + FTC expands the possibility of offering 2-drug simplification in maintenance treatment, especially as both drugs are available as generic formulations. The combination demonstrated non-inferiority in efficacy and safety, as well as a greater improvement in QoL over time compared to standard regimens. An ongoing assessment of data from the SIMPL'HIV study regarding healthcare-related costs and patient choices and satisfaction will provide additional insights into potential advantages of dual therapy and PCM.

## Supporting information

**S1 Fig. Study flowchart.**
(TIF)

**S1 Table. Proportion of participants with HIV-1 RNA < 100 copies/ml throughout 48 weeks and FDA snapshot of proportion of patients with HIV-1 RNA < 50 copies/ml at week 48 by monitoring arm.** ITT analysis.
(DOCX)

**S2 Table. Participants experiencing virological failure (according to the outcome definition).**
(DOCX)

**S3 Table. Proportions of patients with reported adverse events by monitoring arm.** ITT analysis.
(DOCX)

**S4 Table. List of reported serious adverse events through 48 weeks.**
(DOCX)

**S1 Text. CONSORT checklist.**
(DOC)

**S2 Text. SIMPL'HIV study protocol.**
(PDF)

## Acknowledgments

The authors would like to thank the study participants for their cooperation in this study.

They thank also the following persons for their contributions: Rosemary Sudan and Mikaela Smit (review and editing); Charlotte Barbieux, Tamara Da Silva, Christelle Martin, and Alice Pannerec (investigation, data curation, validation); Christina Grube and Louise Seiler (investigation); Mattia Branca and Armando Lenz (formal analysis); Dominique Rubi (software); and the Clinical Research Center, Geneva University Hospitals and the University of Geneva Faculty of Medicine (validation).

**Membership of the Swiss HIV Cohort Study:** Aebi-Popp K, Anagnostopoulos A, Battegay M, Bernasconi E, Böni J, Braun DL, Bucher HC, Calmy A, Cavassini M, Ciuffi A, Dollenmaier G, Egger M, Elzi L, Fehr J, Fellay J, Furrer H, Fux CA, Günthard HF (President of the SHCS), Haerry D (Deputy of "Positive Council"), Hasse B, Hirsch HH, Hoffmann M, Hösli I, Huber M, Kahlert CR (Chairman of the Mother & Child Substudy), Kaiser L, Keiser O, Klimkait T, Kouyos RD, Kovari H, Ledergerber B, Martinetti G, Martinez de Tejada B, Marzolini C, Metzner KJ, Müller N, Nicca D, Paioni P, Pantaleo G, Perreau M, Rauch A (Chairman of the Scientific Board), Rudin C, Scherrer AU (Head of Data Centre), Schmid P, Speck R, Stöckle M (Chairman of the Clinical and Laboratory Committee), Tarr P, Trkola A, Vernazza P, Wandeler G, Weber R, Yerly S.

## Author Contributions

**Conceptualization:** Delphine Sculier, Gilles Wandeler, Sabine Yerly, Marcel Stoeckle, Enos Bernasconi, Dominique L. Braun, Pietro Vernazza, Matthias Cavassini, Marta Buzzi, Karin J. Metzner, Laurent A. Decosterd, Huldrych F. Günthard, Patrick Schmid, Andreas Limacher, Matthias Egger, Alexandra Calmy.

**Data curation:** Annalisa Marinosci, Marta Buzzi, Andreas Limacher.

**Formal analysis:** Delphine Sculier, Gilles Wandeler, Andreas Limacher.

**Funding acquisition:** Delphine Sculier, Sabine Yerly, Pietro Vernazza, Alexandra Calmy.

**Investigation:** Gilles Wandeler, Sabine Yerly, Annalisa Marinosci, Marcel Stoeckle, Enos Bernasconi, Dominique L. Braun, Pietro Vernazza, Matthias Cavassini, Marta Buzzi, Karin J. Metzner, Laurent A. Decosterd, Patrick Schmid, Alexandra Calmy.

**Methodology:** Delphine Sculier, Gilles Wandeler, Huldrych F. Günthard, Andreas Limacher, Matthias Egger, Alexandra Calmy.

**Project administration:** Annalisa Marinosci, Alexandra Calmy.

**Resources:** Gilles Wandeler, Marcel Stoeckle, Enos Bernasconi, Dominique L. Braun, Pietro Vernazza, Matthias Cavassini, Huldrych F. Günthard, Patrick Schmid, Alexandra Calmy.

**Software:** Andreas Limacher.

**Supervision:** Gilles Wandeler, Andreas Limacher, Alexandra Calmy.

**Validation:** Delphine Sculier, Gilles Wandeler, Sabine Yerly, Annalisa Marinosci, Marta Buzzi, Andreas Limacher, Alexandra Calmy.

**Visualization:** Annalisa Marinosci, Andreas Limacher.

**Writing – original draft:** Delphine Sculier, Annalisa Marinosci, Alexandra Calmy.

**Writing – review & editing:** Delphine Sculier, Sabine Yerly, Annalisa Marinosci, Marcel Stoeckle, Enos Bernasconi, Dominique L. Braun, Pietro Vernazza, Matthias Cavassini,

Marta Buzzi, Karin J. Metzner, Laurent A. Decosterd, Huldrych F. Günthard, Patrick Schmid, Andreas Limacher, Matthias Egger, Alexandra Calmy.

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
