## [Editor Report · Decision Letter 0]

13 Mar 2020

Dear Dr Sculier, 

Thank you for submitting your manuscript entitled "Efficacy and safety of dolutegravir plus emtricitabine versus standard ART for the maintenance of HIV-1 suppression: 48-week results of the factorial, randomized, non-inferiority SIMPL-HIV trial" for consideration at PLOS Medicine.

Your manuscript has now been evaluated by the PLOS Medicine editorial staff, as well as by an academic editor with relevant expertise, and I am writing to let you know that we would like to send your submission out for external review.

Kind regards,

Richard Turner, PhD

Senior editor, PLOS Medicine

rturner@plos.org

---

## [Decision Letter · Decision Letter 1]

25 Aug 2020

Dear Dr. Sculier,

Thank you very much for submitting your manuscript "Efficacy and safety of dolutegravir plus emtricitabine versus standard ART for the maintenance of HIV-1 suppression: 48-week results of the factorial, randomized, non-inferiority SIMPL-HIV trial" (PMEDICINE-D-20-00718R1) for consideration at PLOS Medicine. We do apologize for the delay in sending you a decision. 

Your paper was evaluated by an academic editor with relevant expertise and sent to independent reviewers, including a statistical reviewer. The reviews are appended at the bottom of this email and any accompanying reviewer attachments can be seen via the link below:

[LINK]

In light of these reviews, we will not be able to accept the manuscript for publication in the journal in its current form, but we would like to invite you to submit a revised version that addresses the reviewers' and editors' comments fully. You will appreciate that we cannot make a decision about publication until we have seen the revised manuscript and your response, and we expect to seek re-review by one or more of the reviewers. 

We hope to receive your revised manuscript by Sep 15 2020 11:59PM. Please email us (plosmedicine@plos.org) if you have any questions or concerns.

Please let me know if you have any questions, and otherwise we will look forward to receiving your revised manuscript in due course. 

Sincerely,

Richard Turner, PhD

rturner@plos.org

Please quote the dates of start and end of study recruitment in the abstract, and participants' median age.

Please add a new final sentence to the "Methods and findings" subsection of the abstract, quoting 2-3 of the study's main limitations. 

At line 102, please start the sentence "In this study ...".

After your abstract, we will need to ask you to add a new and accessible "author summary" section in non-identical prose. You may find it helpful to consult one or two recent research papers published in PLOS Medicine to get a sense of the preferred style. 

At line 299 and any other instances, please use "White" rather than "Caucasian".

Please avoid claims of "the first", e.g. at line 408, and where needed add "to our knowledge" or similar. 

Throughout the text, please quote p values alongside 95% CI, where available. To clarify presentation of 95% CI, we suggest using "to" to indicate the range, rather than an em rule (e.g., "-7.8 to 5.6").

Please convert reference call-outs to the following style throughout the text: "... maintenance therapy [1,2]." (noting the missing full point at line 113). 

The details for some references seem incomplete, e.g., reference 16.

Please remove all instances of "[Internet]" from the reference list, and convert French text to English. 

Please include a completed CONSORT checklist as a supplementary file, referred to in the Methods section (e.g., "See S1_CONSORT_Checklist"). In the checklist, please refer to individual items by section (e.g., "Methods") and paragraph number rather than by line or page numbers, as the latter generally change in the event of publication. 

Comments from the reviewers:

*** Reviewer #1: 

Statistical review

This paper reports a randomised controlled trial comparing continuing cART with switching to dual DTG+FTC therapy in HIV patients who are virally supressed. A secondary randomisation is between a simplified monitoring strategy and normal monitoring. The authors show the DTG+FTC therapy is non-inferior to continuing cART, with some QoL advantages. Generally the paper used appropriate statistical methods. I have some minor comments on the reporting:

1. Abstract "patients were randomized 1:1:1:1…" I would recommend rewriting this sentence to emphasise that patients were randomised 1:1 between continuing cART and switching, and 1:1 between PCM and SM.

2. Abstract line 92/93 - I would recommend putting the non-inferiority margin earlier in the abstract and giving a p-value for non-inferiority in addition to the CI for risk difference.

3. Methods - there is no mention of blinding. the study was not double-blinded, but was there any blinding (e.g. rater-blinding?).

4. Methods, outcomes - I would recommend all secondary outcomes listed on the clinicaltrials.gov registration are mentioned in the main body of the paper (even if they are not reported in the paper).

5. Methods, sample size calculation - "We primarily analyzed the trial as a stratified two-arm trial" - I would mention the stratification factor in this sentence (i.e. the monitoring type).

6. Methods, sample size calculation - was the sample size chosen to find a difference in monitoring types, if it existed? I see in the protocol this is based on cost differences, which are not reported in this paper - I would suggest it might be informativ to add this to the sample size calculation and say that these outcomes are not reported in this paper.

7. Results line 287 - I didn't follow what a randomisation occurring by mistake meant - was the patient ineligible? 

8. Results - I would recommend all secondary endpoints are reported in a table in the main body of the paper, as opposed to in supplementary material.

9. Line 394 - I recommend adding 'significant' between 'no' and 'difference'.

James Wason

*** Reviewer #2: 

Overall a well-designed and conducted study demonstrating the non-inferiority of DTG/FTC to standard cART in maintaining virologic suppression in treated patients. 

Minor comments: 

1. More needs to be explicitly said regarding the relatively low number of female participants recruited to the study, in comparison to the gender proportions of the Swiss HIV cohort as a whole. 

2. In general, the authors can consider making more explicit the rationale for the entire study - it is not immediately clear why the combination of DTG/FTC is required given that the evidence for DTG/3TC is fairly well-established, and FTC and 3TC are essentially clinically equivalent. A statement on this would clarify their position, as well as situate the study in the context of antiretroviral choices available to patients and prescribers. 

3. It is somewhat surprising that there was no significant difference in the QOL indices between participants undergoing SCM and PCM. What are some of the postulated reasons for this? 

Otherwise, no other issues and would recommend acceptance. 

*** Reviewer #3: 

Overall, this is a well-executed trial with appropriate study design. The biggest aspect that needs to be addressed in this study is why it was conducted, how it will supplement the data already available, and in what patient scenarios would this data be applicable. Additionally, one of the secondary outcomes is quality of life, with this trial being an open-label design you allow for bias while assessing subjective matters. Generally, I felt the article verbiage flowed well and there were few grammatical errors. I did feel that the difference between the patient-centered simplified monitoring (PCM) and the standard monitoring (SM) could have been elaborated on further. 

Global Recommendations: 

1. Instead of "HIV-1 infected adults" use "people living with HIV (PLWH)"

Abstract: 

1. On page 4, they define virologic suppression as having < 100 copies/mL. Later on page 7, under the methods section, they state that they "define virological suppression as HIV-11 RNA of < 50 copies/mL"

Introduction:

1. Need more convincing of why this research was conducted? The pill burden would be increasing for the patient which could lead to decreased adherence and worse clinical outcomes. Why would you ever make this switch in clinical practice when there is dolutegravir/lamivudine? 

2. On page 7, in the introduction section, there is a statement that says "As a second primary objective, the study aimed to compare the costs of the patient-centered monitoring (PCM) approach with standard 3-monthly routine surveillance" - within the primary outcome section, there is no mention of the results of this objective. It is never addressed.

Methods: 

1. On page 4, they define virologic suppression as having < 100 copies/mL. Later on page 7, under the methods section, they state that they "define virological suppression as HIV-11 RNA of < 50 copies/mL"

 Results:

1. Table 1, the demographic between the study group and they cART group. Based on the patient information on the table, the DTG+FTC group appears to be more compromised than the cART group, The DTG+FTC patient's median CD4 baseline is low, while the medical viral load was higher. Additionally, the DTG+FTC group had been receiving ART longer than the cART group, on average. 

2. An aspect I found interesting was that if you look at table 3 on page 17, there appear to be substantially more serious adverse effects from the cART group compared to the DTG+FTC group. This was not highlighted in the article. 

3. On page 17, they point out that the CrCl for the DTG+FTC group is significantly lower than the cART patients. The authors do not offer any explanation of why they believe this to be happening.

***

[LINK]

---

## [Editor Report · Decision Letter 2]

25 Sep 2020

Dear Dr. Sculier,

Thank you very much for re-submitting your manuscript "Efficacy and safety of dolutegravir plus emtricitabine versus standard ART for the maintenance of HIV-1 suppression: 48-week results of the factorial, randomized, non-inferiority SIMPL-HIV trial" (PMEDICINE-D-20-00718R2) for consideration at PLOS Medicine.

I have discussed the paper with editorial colleagues and our academic editor, and I am pleased to tell you that, provided the remaining editorial and production issues are fully dealt with, we expect to be able to accept the paper for publication in the journal.

The issues that need to be addressed are listed at the end of this email. Any accompanying reviewer attachments can be seen via the link below. Please take these into account before resubmitting your manuscript:

[LINK]

Please let me know if you have any questions. Otherwise, we look forward to receiving the revised manuscript shortly. 

Sincerely,

Richard Turner, PhD

rturner@plos.org

Requests from Editors:

We note that your data statement describes all relevant data as being available in the article and supplementary files. To comply with PLOS' data policy please include patient-level data in supplementary files, or arrange to make these available at a public repository, say (https://journals.plos.org/plosmedicine/s/data-availability).

In your abstract, rather than "Noninferiority margin ... was -12%", please adapt the explanation to match that around line 330, i.e., "noninferiority was declared if the lower bound of the 95% confidence interval was greater than -12%" or similar. 

Please mention both ITT and per protocol analyses in the abstract, e.g., by specifying that the primary analysis used ITT (assuming this is correct) and noting that the results of a per protocol analysis were similar. 

Please add a sentence, say, to the abstract to provide additional information on adverse events. We would suggest quoting the higher incidence of serious adverse events in the cART arm, and noting that events were not judged to be related to study medication. 

Pleas reformat the author summary with bullet points, aiming for around 3 points in each of the three subsections. Please trim the individual points, which should consist of no more than 1-2 short sentences each. 

Please correct "jeopardizing" in the author summary. 

In the methods section, we notice that the noninferiority criterion is discussed around lines 313 and 330. We suggest avoiding repetition. 

Please change "gender" to "sex" where appropriate, e.g., at line 536.

Throughout the text, please format reference call-outs as follows: "... clinical trials [9,10].".

Please review the reference list carefully. In some cases the journal names need to be adapted, e.g., "Lancet" will suffice for reference 35; and all italics should be removed, e.g., from reference 36. 

Noting reference 8, please ensure that all references contain full access information. 

Please remove all elements of French spelling, e.g. "Jan" rather than "janv" for reference 31. 

Please remove all examples of "[Internet]", e.g., from reference 30. Please also spell out the institutional author name for this reference. 

Please remove the page numbers from the attached CONSORT checklist - individual items should be referred to by article section, as at present, and paragraph number.

***

---

## [Editor Report · Decision Letter 3]

14 Oct 2020

Dear Dr Sculier, 

On behalf of my colleagues and the academic editor, Dr. Marie-Louise Newell, I am delighted to inform you that your manuscript entitled "Efficacy and safety of dolutegravir plus emtricitabine versus standard ART for the maintenance of HIV-1 suppression: 48-week results of the factorial, randomized, non-inferiority SIMPL-HIV trial" (PMEDICINE-D-20-00718R3) has been accepted for publication in PLOS Medicine. 

PRODUCTION PROCESS

Before publication you will see the copyedited word document (within 5 busines days) and a PDF proof shortly after that. The copyeditor will be in touch shortly before sending you the copyedited Word document. We will make some revisions at copyediting stage to conform to our general style, and for clarification. When you receive this version you should check and revise it very carefully, including figures, tables, references, and supporting information, because corrections at the next stage (proofs) will be strictly limited to (1) errors in author names or affiliations, (2) errors of scientific fact that would cause misunderstandings to readers, and (3) printer's (introduced) errors. Please return the copyedited file within 2 business days in order to ensure timely delivery of the PDF proof. 

If you are likely to be away when either this document or the proof is sent, please ensure we have contact information of a second person, as we will need you to respond quickly at each point. Given the disruptions resulting from the ongoing COVID-19 pandemic, there may be delays in the production process. We apologise in advance for any inconvenience caused and will do our best to minimize impact as far as possible.

PRESS

PROFILE INFORMATION

Thank you again for submitting the manuscript to PLOS Medicine. We look forward to publishing it. 

Best wishes, 

Richard Turner, PhD

Senior Editor 

PLOS Medicine

plosmedicine.org